Extended Abstract Track

# Category-Level 6D Object Pose Estimation in the Wild: A Semi-Supervised Learning Approach and A New Dataset

**Editors:** List of editors' names

## Abstract

6D object pose estimation is one of the fundamental problems in computer vision and robotics research. While a lot of recent efforts have been made on generalizing pose estimation to novel object instances within the same category, namely category-level 6D pose estimation, it is still restricted in constrained environments given the limited number of annotated data. In this paper, we collect Wild6D, a new unlabeled RGBD object video dataset with diverse instances and backgrounds. We utilize this data to generalize category-level 6D object pose estimation in the wild with semi-supervised learning. We propose a new model, called **Re**ndering for **Po**se estimation network (**RePoNet**), that is jointly trained using the free ground-truths with the synthetic data, and a silhouette matching objective function on the real-world data. Without using any 3D annotations on real data, our method outperforms state-of-the-art methods on the previous dataset and our Wild6D test set (with manual annotations for evaluation) by a large margin. Our code and dataset will be made publicly available.

## 1. Introduction

Estimating the 6D object pose is one of the core problems in computer vision and robotics. It predicts the full configurations of rotation, translation and size of a given object, which has wide applications including Virtual Reality (VR) (Biocca, 1992), scene understanding (Marder-Eppstein, 2016), and  (Tremblay et al., 2018; Zhu et al., 2014; Mousavian et al., 2019; Wen et al., 2020). There are two directions in 6D object pose estimation. One is performing instance-level 6D pose estimation, where a model is trained to estimate the pose of one exact instance with an existing 3D model (He et al., 2020; Peng et al., 2019; Li et al., 2018; Xiang et al., 2017; Oberweger et al., 2018; Chen et al., 2020b; He et al., 2021). However, learning instance-level model restricts its generalization ability to unseen objects. To achieve generalization to unseen instance, another direction is recently proposed to perform category-level 6D pose estimation using one model (Wang et al., 2019; Chen et al., 2020a; Tian et al., 2020; Manhardt et al., 2020; Chen et al., 2021). However, the large appearance and shape variance across instances largely increase the difficulty in learning.

To overcome this limitation, Wang et al. (2019) take the initial step to collect real-world dataset and annotations for category-level 6D pose estimation. Combining synthetic data with free ground-truth annotations, they show the learned model can be generalized to unseen objects within the same category. While this result is encouraging, the generalization ability of the model is still limited by the number and the diversity of the data due to the challenges in annotating 6D object poses. Specifically, only 8,000 images across 13 scenes are collected and annotated from the dataset proposed in (Wang et al., 2019). Thus it is still very challenging to generalize 6D pose estimation on diverse objects in complex scenes.

Training    Testing results on Wild6D

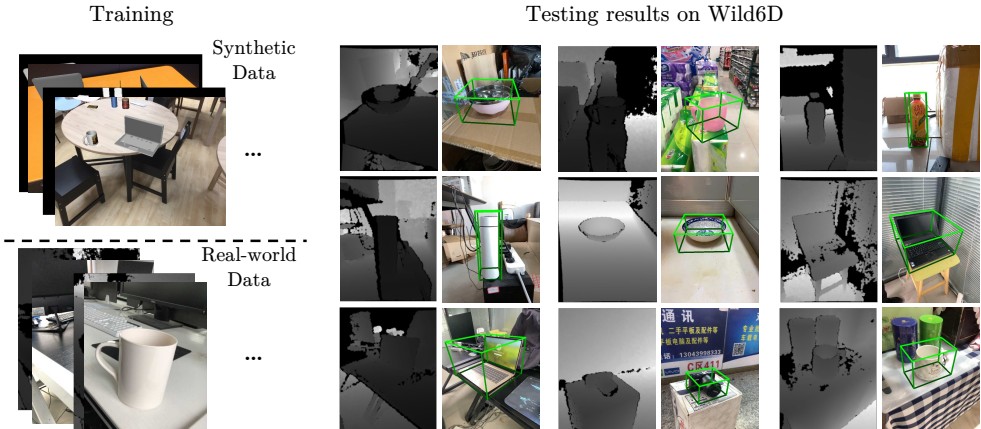

Figure 1: **Left**: we train our model with both synthetic data and real-world data under our proposed semi-supervised setting. **Right**: during inference, given the RGBD images, the object pose can be estimated precisely. Green bounding boxes show the 3D bounding boxes projection results on 2D images.

In this paper, we propose to generalize category-level 6D object pose estimation in the wild. To achieve this goal, we introduce a new dataset and a new semi-supervised learning approach. Our key insight is that, while annotating the 6D object pose is challenging, collecting the RGBD videos for these objects without labels is much easier and more affordable. On the other hand, there are infinite ground-truth annotations for synthetic data which come for free. We propose to leverage the benefits from both sides. We first collect a rich object-centric video dataset with diverse backgrounds and object instances using an RGBD camera, namely *Wild6D*. Each video in the dataset shows multiple views of one or multiple objects (see examples in Fig. 1). In total, there are 5166 videos (> 1.1 million images) over 1722 object instances and 5 categories, which is significantly (300x) larger than the previous 6D object pose estimation dataset (Wang et al., 2019). For evaluation, we annotate 486 videos over 162 objects.

Given this new dataset, we further design a novel model for semi-supervised learning, called **Re**ndering for **Po**se estimation **Net**work (RePoNet). The RePoNet is composed of two branches of networks with a *Pose Network* to estimate the 6D object pose and a *Shape Network* to estimate the 3D object shape. Given the estimated pose and object shape, we perform differentiable rendering to obtain an object mask projected in 2D plane. We train our model jointly using synthetic data with the free ground-truth 6D pose annotations and the unlabeled real-world RGBD videos via a silhouette matching objective. In this way, our pose estimation model can be generalized to in-the-wild data with minimum human labor.

## 2. Wild6D Dataset

**Wild6D Collection.** To achieve a categorical 6D pose for real objects, we collect a new large-scale RGBD dataset, named *Wild6D*. Each video in the Wild6D is recorded via the iPhone front camera showing multiple views of objects where RGB images and the corresponding depth images and point cloud are captured simultaneously. The videos are captured by different turkers with their own iPhones to guarantee the diversity of instances

Extended Abstract Track

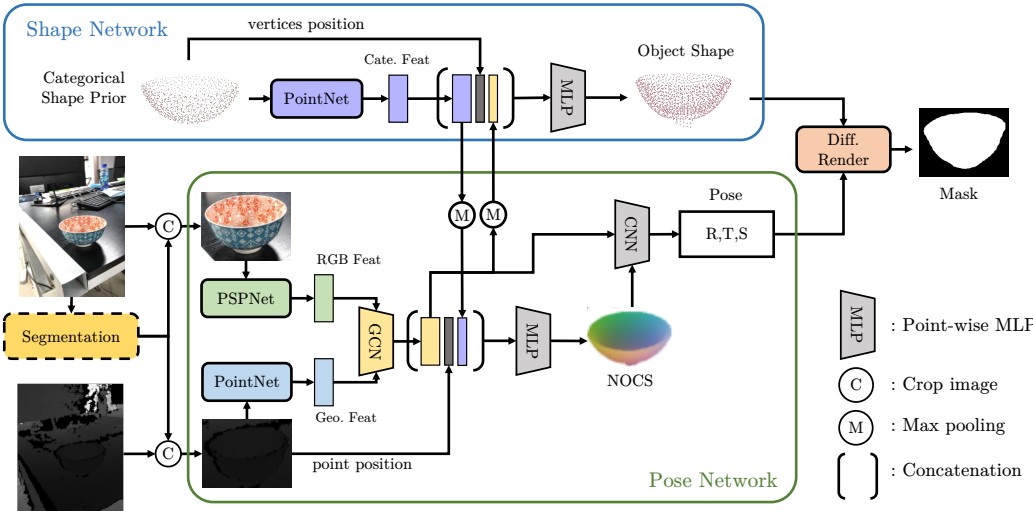

Figure 2: Overview of the proposed method. Given the input image and depth map, RePoNet estimates the object pose, NOCS map, and shape simultaneously via Pose Network and Shape Network. These two networks are bridged via the differentiable rendering module.

and background scenes. Three videos are taken for each object under different scenes. In total, Wild6D consists of 5,166 videos (>1.1 million images) over 1722 different object instances and 5 categories , *i.e.*, *bottle*, *bowl*, *camera*, *laptop*, and *mug*. Among this data, we split 486 videos of 162 instances to use them as the test set. Our proposed Wild6D data significantly improves the number of images, object instances, and scene complexity compared with the existing datasets.

**Wild6D Annotation.** To annotate more than 10,000 images in the testing set efficiently, we propose a tracking-based annotation pipeline. Inside a video, we manually annotate the 6D object poses every 50 frames as keyframes. Given the annotation of the keyframe, we implement TEASER++ (Yang et al., 2020) together with colored ICP (Park et al., 2017) to achieve the registration between the keyframe and the following frame and compute the transformation matrix. The ground-truth object pose of the following frame can be obtained by applying the transformation matrix to the keyframe annotation. Following this pipeline, we can obtain accurate ground-truths by only labeling around 5 keyframes per video.

## 3. Proposed Method

**RePoNet overview.** As illustrated in Fig. 2, the RePoNet is composed of two networks including the *Pose Network* to directly estimate the object 6D pose parameters (with the NOCS map as an intermediate representation) and the *Shape Network* to reconstruct the object shape. The outputs from both networks can go through a differentiable rendering (Liu et al., 2019) module which outputs a segmentation mask.

**Semi-supervised learning setting**. We utilize two sets of data for semi-supervised learning: (i) synthetic data with full annotations including NOCS maps, CAD models, foreground segmentation, and 6D pose parameters, denoted as $\mathbf{D}_{syn}$; and (ii) a large-scale real-world RGBD dataset Wild6D with estimated foreground masks (using pre-trained Mask R-CNN He et al. (2017)), denoted as $\mathbf{D}_{real}$. Both the synthetic data and the real-world data

| Methods | IOU$_{0.5}$ | 5 degree 2cm | 5 degree 5cm | 10 degree 5cm |
|---|---|---|---|---|
| NOCS (Wang et al., 2019) | 78.0 | 7.2 | 10.0 | 25.2 |
| Shape-Prior (Tian et al., 2020) | 77.3 | 19.3 | 21.4 | 54.1 |
| FS-Net (Chen et al., 2021) | **92.2** | – | 28.2 | 60.8 |
| DualPoseNet (Lin et al., 2021) | 79.8 | 29.3 | 35.9 | 66.8 |
| SGPA (Chen and Dou, 2021) | 80.1 | **35.9** | 39.6 | 70.7 |
| GPV-Pose Di et al. (2022) | 83.0 | 32.0 | **42.9** | **73.3** |
| RePoNet-sup | 81.1 | 35.1 | 40.4 | 68.8 |
| CPS++ (Manhardt et al., 2020) w/ ICP | 72.8 | – | 25.2 | ≤ 58.6 |
| SSC-6D Peng et al. (2022) w/ ICP | 72.7 | 28.6 | 33.4 | 62.9 |
| CPPF You et al. (2022) | 26.4 | – | 16.9 | 44.9 |
| RePoNet-semi | **76.0** | **30.7** | **33.9** | **63.0** |

Table 1: **Comparison of our approach with the SOTA methods on REAL275**. The best results are highlighted in bold.

| Methods | IOU$_{0.5}$ | 5 degree 2cm | 5 degree 5cm | 10 degree 5cm |
|---|---|---|---|---|
| CASS (Chen et al., 2020a) | 1.04 | 0 | 0 | 0 |
| Shape-Prior (Tian et al., 2020) | 32.5 | 2.6 | 3.5 | 13.9 |
| DualPoseNet (Lin et al., 2021) | 70.0 | 17.8 | 22.8 | 36.5 |
| GPV-Pose (Di et al., 2022) | 67.8 | 14.1 | 21.5 | 41.1 |
| RePoNet-syn | 66.7 | 26.0 | 30.8 | 40.3 |
| RePoNet-semi | **70.3** | **29.5** | **34.4** | **42.5** |

Table 2: **Comparison of our approach with the SOTA methods on Wild6D.** The best results are highlighted in bold.

are utilized to train our RePoNet jointly. For the synthetic data, all the ground-truths are used as supervision signals for the *Pose Network* and the *Shape Network*. While for the real-world data, we directly feed the outputs from the *Pose Network* and the *Shape Network* into the differentiable rendering module to generate the binary mask, then train the whole RePoNet by comparing the rendered mask with the object foreground mask (silhouette matching loss). While there can be a more delicate way to deal with the domain gap between simulation and real, we find the direct sharing of parameters during training already provides a simple yet effective solution for generalization. One important contribution of this framework is to make the whole procedure with RePoNet differentiable, using the Shape Network, and a ConvNet to connect NOCS map to 6D pose parameters in the Pose Network. This allows the gradients to backprop through the network end-to-end.

## 4. Experiments

We conduct experiments on both NOCS REAL275 dataset and the proposed Wild6D dataset. **Results on REAL275**. We split all existing methods into two groups by whether using full annotations of real data and compare their performance with RePoNet under both fully-supervised setting and semi-supervised setting, denoted as "RePoNet-sup" and "RePoNet-semi" respectively as listed in Table 1.
**Results on Wild6D**. We directly test the models pretrained on NOCS (Wang et al., 2019) on Wild6D test set and report the performance in Table 2. The significant improvement over other state-of-the-art shows the better generalization ability of RePoNet. Also, the superior performance of RePoNet-semi over RePoNet-syn shows our proposed semi-supervised training can effectively leverage the in-the-wild data.

# Extended Abstract Track

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

## Appendix A. First Appendix

This is the first appendix.

## Appendix B. Second Appendix

This is the second appendix.

