# OpenReview forum: "Category-Level 6D Object Pose Estimation in the Wild: A Semi-Supervised Learning Approach and A New Dataset"
_NeurIPS.cc/2022/Workshop/NeurReps — NeurReps 2022 Poster_

### Official Review · Reviewer_FbN7 · 2022-10-05

**Confidence:** 4
**Soundness:** 3
**Presentation:** 3
**Contribution:** 4
**Overall Rating:** 6

**Summary:**

This work investigates 6D object pose detection in the wild.  That is, they focus on the setting where there is a rich variety of object instances and backgrounds, without access to annotated labels.  Their contributions are two-fold.  First, they propose a new architecture, RePoNet, which can be trained with either full ground-truth annotations or foreground masks.  The foreground masks can be generated in easily with an off-the-shelf segmentation network, allowing RePoNet to train on real-world data without full annotations.  Second, they introduce a new dataset, Wild6D, which includes RGBD video sequences of many object instances in different scenes, to serve as a benchmark for pose detection in the wild.

**Questions:**

- Why does RePoNet do worse than other methods in the supervised case?  It is not clear why the additional ability to generate silhouettes, would hurt performance when training on full annotations.
- How is rotational symmetry handled in the Wild6D annotations?

**Limitations:**

It would be helpful to provide a table comparing the proposed Wild6D dataset to existing 6D object pose detection datasets in terms of: number of images, number of object instances, type of annotation, etc.

**Recommended Decision:**

3: Accept

**Relevance:**

3: Solid fit

**Strengths And Weaknesses:**

Strengths:
- The authors identify an important limitation in existing 6D object pose detection research: current datasets do not cover a diverse set of objects or scenes.  The proposed Wild6D expands the number of object instances greatly and could serve as a valuable benchmark to test generalization abilities.
- The authors propose a novel method for training on real-world data with a silhouette matching loss.  Moreover, the differentiable rendering used to produce the silhouettes takes as input the object shape and transformation parameters, which can also be trained in a supervised manner when labels are available.  They show empirically that the silhouette matching loss improves accuracy on the Wild6D dataset.

Weaknesses:
- The authors discuss using an automated framework to generate labels between manually annotated keyframes.  It would be helpful to evaluate the accuracy of these annotations, perhaps by comparing to held-out manual annotations.
- The components of RePoNet were not discussed in enough detail.  Where does the categorical shape prior come from? How does the segmentation work?
- The authors did not discuss the limitations of the silhouette matching approach.  For instance, it seems that such a loss function would be weak when training on symmetric objects like a bowl.


**Submission Track:**

Extended Abstract (4 Page)

---

### Official Review · Reviewer_dq33 · 2022-10-14

**Confidence:** 4
**Soundness:** 3
**Presentation:** 2
**Contribution:** 4
**Overall Rating:** 6

**Summary:**

The authors collect a new dataset of indoor object-centric videos for the purpose of real-world 6D pose estimation. A subset of the videos is labeled for evaluation. In addition, a novel semi-supervised methods called RePoNet is proposed. The method combines supervised training on synthetic data and self-supervised training on real-world data. The self-supervised objective is based on shape priors and differentiable rendering.

**Questions:**

* Do you always predict the pose of a single object in your new dataset? How does the model know which of the objects in the image it should be making a prediction for?
* When will you make your dataset publicly available?

**Limitations:**

The proposed method is not fully described. Neither are its limitations.

**Recommended Decision:**

3: Accept

**Relevance:**

3: Solid fit

**Strengths And Weaknesses:**

Overall, the contribution is very strong. The authors collect a new real-world pose regression dataset and propose a method that can achieve strong results on real-world benchmarks using a combination of supervised and self-supervised. Leveraging the plethora of real-world unlabeled data is an important direction in machine learning.

What is missing is a proper description of the method (many aspects of Figure 2 are not described in the text) and a description of the metrics used for evaluation. The unfinished appendix suggest that this paper was put together in a hurry. Please complete the description for the camera-ready version.

Minor comments:
* “Short Title” at the top of page 3.
* “We split all existing methods into two groups by whether using full annotations of real data “ – re-word
* The acronym "NOCS" is used without any explanation.

**Submission Track:**

Extended Abstract (4 Page)

---

### Official Review · Reviewer_upXs · 2022-10-18
**Interesting paper, good results, could have been written better**

**Confidence:** 4
**Soundness:** 3
**Presentation:** 2
**Contribution:** 3
**Overall Rating:** 5

**Summary:**

The paper addresses the problem of 6D pose estimation from RGBD images in the wild. The contribution is twofold: a novel dataset, named Wild6D, and a novel method, named RePoNet.
Wild6D contains more than 5K videos capturing 1.7K object instances from 5 categories collected using Amazon Mechanical Turk. Pros: The proposed dataset is significantly larger and contains more diverse backgrounds and object instances than the competitor NOCS dataset proposed by Wang et al. 2019. Only a small portion of the dataset is annotated.
RePoNet is an end-to-end deep architecture containing two branches: a pose network to estimate the 6D object pose using the NOCS map as an intermediate representation, and a shape network to reconstruct the 3D object shape by deforming a categorical shape prior. Training is performed jointly on both annotated synthetic data and unlabeled real data. When training on the former the output of the pose and shape networks are compared with the ground-truth annotations. When training on the latter a differentiable rendering is used to obtain an object mask projected to the 2D plane and asilhouette matching loss is used to enforce consistency.

**Questions:**

Some modules in Figure 2 are not described, e.g. PSPNet and GCN. What is the GCN module? I guess it stands for Graph Convolutional Network but more details are needed to understand how it is applied, the same holds for PSPNet.
What are the metrics reported in Table 1 and Table 2 for the rightmost three columns? Is the IoU computed between 3D bounding boxes or between their 2D projections?
In the semi supervised case, which percentage of annotations is dropped?

Typos / Minor issues:
In the abstract you wrote "it is still restricted in constrained environments." I don't understand what do you mean.
Missing term: VR, scene understanding, and ?
In the introduction you wrote: "another direction is to perform category-level 6D pose estimation using one model." What do you mean by "using one model"?
Why is there a minor or equal symbol in the in the last column of the CPS++ method in Table 1?

**Limitations:**

Limitations are not described.

**Recommended Decision:**

3: Accept

**Relevance:**

3: Solid fit

**Strengths And Weaknesses:**

Strengths
Wild6D dataset: The proposed dataset is significantly larger and contains more diverse backgrounds and object instances than the competitor NOCS dataset proposed by Wang et al. 2019.
RePoNet: The network design is quite interesting, learns the 6D pose in a end-to-end fashion, and achieves compelling results in the semi-supervised setting, gaining a consistent margin against competitor methods even without using an ICP refinement as postprocessing.

Weaknesses
The paper could have written better. Due to the limited space I found the amount of repetitions in the first part excessive. Important details are missing (see Questions).

**Submission Track:**

Extended Abstract (4 Page)

---

### Decision · Program_Chairs · 2022-10-21

Accept (Poster)